# Thyroidal and Extrathyroidal Requirements for Iodine and Selenium: A Combined Evolutionary and (Patho)Physiological Approach

**DOI:** 10.3390/nu14193886

**Published:** 2022-09-20

**Authors:** D. A. Janneke Dijck-Brouwer, Frits A. J. Muskiet, Richard H. Verheesen, Gertjan Schaafsma, Anne Schaafsma, Jan M. W. Geurts

**Affiliations:** 1University of Groningen, University Medical Center Groningen, Department of Laboratory Medicine, Hanzeplein 1, 9713 GZ Groningen, The Netherlands; 2Regionaal Reuma Centrum Z.O. Brabant Máxima Medisch Centrum, Ds. Th. Fliednerstraat 1, 5631 BM Eindhoven, The Netherlands; 3Schaafsma Advisory Services in Food, Health and Safety, Rembrandtlaan 12, 3925 VD Scherpenzeel, The Netherlands; 4FrieslandCampina, 3818 LE Amersfoort, The Netherlands

**Keywords:** iodine, selenium, evolution, dietary reference intakes, thyroid, peroxidase partner system, exocrine glands, autoimmune thyroid disease, cancer, seafood

## Abstract

Iodide is an antioxidant, oxidant and thyroid hormone constituent. Selenoproteins are needed for triiodothyronine synthesis, its deactivation and iodine release. They also protect thyroidal and extrathyroidal tissues from hydrogen peroxide used in the ‘peroxidase partner system’. This system produces thyroid hormone and reactive iodine in exocrine glands to kill microbes. Exocrine glands recycle iodine and with high urinary clearance require constant dietary supply, unlike the thyroid. Disbalanced iodine-selenium explains relations between thyroid autoimmune disease (TAD) and cancer of thyroid and exocrine organs, notably stomach, breast, and prostate. Seafood is iodine unconstrained, but selenium constrained. Terrestrial food contains little iodine while selenium ranges from highly deficient to highly toxic. Iodine vs. TAD is U-shaped, but only low selenium relates to TAD. Oxidative stress from low selenium, and infection from disbalanced iodine-selenium, may generate cancer of thyroid and exocrine glands. Traditional Japanese diet resembles our ancient seashore-based diet and relates to aforementioned diseases. Adequate iodine might be in the milligram range but is toxic at low selenium. Optimal selenoprotein-P at 105 µg selenium/day agrees with Japanese intakes. Selenium upper limit may remain at 300–400 µg/day. Seafood combines iodine, selenium and other critical nutrients. It brings us back to the seashore diet that made us what we currently still are.

## 1. Introduction

Iodine and selenium are two interacting elements. They are important for (brain) growth and metabolism and protect against free radicals and pathogenic microbes. While the role of iodine in the formation of thyroid hormones is well established, the role of selenium receives less attention. Selenoproteins play a role in thyroid hormone metabolism and the liberation of iodide from organic stores. They also protect thyrocytes against the hydrogen peroxide (H_2_O_2_) that is abundantly generated for thyroid hormone synthesis. Apart from its role in thyroid hormone synthesis, iodide is probably the most ancient antioxidant used by organisms for protection against oxidative stress. Because of this property, seaweed became famous for its high iodine content, but its appreciable selenium content is less known. In a similar reaction both elements are part of the so-called iodide/hydrogen-peroxide/peroxidase partner system, hereafter named the ‘peroxidase partner system’. This system makes use of the combination of iodide and H_2_O_2_ as an oxidant and can be found in virtually every exocrine gland. It is used as a defense system against microorganisms that are close to the lining of their lumen. Accordingly, iodine deficiency and disbalances between iodine and selenium are intimately related to impaired (brain) growth and development in infancy, but also to thyroid autoimmune disease and thyroid cancer, as well as cancer of the exocrine glands, such as malignancies of the female breast, prostate and stomach in adults.

In this paper, we argue that current recommendations of iodine are too low and in need of revision, taking the protecting effect of selenium into account as well as iodine’s high intakes in human evolution and still in many Asian countries. The current recommendations of selenium find a solid scientific basis in the optimization of selenoprotein levels and in studies showing its toxicity at high intakes. However, positive longer-term effects of selenium intakes above current recommendations should not be ignored. Approaching the subject from a combined evolutionary and mechanistic angle may provide better insight in the short- and longer-term requirements for both elements. In such an evolutionary line of reasoning, we would like to make the case that dietary reference intakes should comply with the ‘whole of the evidence’, which entails not only the most recent meta-analyses of randomized controlled trials (RCTs) but also considers insights from underlying mechanisms and insights from human evolution.

## 2. Physical and (Bio)Chemical Properties of Iodine and Selenium

The properties and metabolism of iodine and selenium are crucial to the understanding of why nature has chosen these elements to take center stage.

Iodine (^127^I) is a halogen in IUPAC group 17 (VIIa), period 5, of the periodic table. It is an element with 53 protons, 74 neutrons and 37 known isotopes, of which 2 occur naturally: the stable ^127^I (virtually 100%) and a trace of ^129^I with a half-life of 1.57 × 10^7^ years. The synthetic ^123^I, ^124^I and notably the gamma-emitter ^125^I (long term tracking/imaging; half-life 60 days) and the beta-emitter ^131^I (treatment; half-life 8 days) are used as tracers and therapeutic agents in medicine [1,2]. The atomic weight is 126.90 and the electron configuration is [Kr] 4d^10^5s^2^5p^5^. Iodine is a nonmetallic, dark gray/purple-black, lustrous, solid element that sublimes easily on heating to give a purple vapor. The oxidation/reduction states are −1, +1, +3, +4, +5, +6, +7 (a strongly acidic oxide), with a rather strong tendency to fill the outer p-orbit with a single electron to mimic krypton and thereby become iodide (I^−^). Among the electronegative halogens, iodine is nevertheless the most electropositive and the least reactive, although it can still associate with many other elements (like carbon in thyroid hormone and oxygen in hypoiodous acid; HOI) and itself (I_2_; molecular iodine). The inorganic iodine species that exist within a human body at pH 7.4 are: iodide (I^−^), molecular iodine (I_2_), triiodide (I_3_), hypoiodous acid (HIO), hypoiodite ion (OI^−^), and the iodine anion (HI_2_O^−^) [3,4].

Being the heaviest atom, carrying the largest number of protons and electrons in our body [3], iodine’s electrons in the outer p-orbit have a ‘loose’ nature. This characteristic, makes iodine useful to function as both electron donor (antioxidant) and electron acceptor (oxidant). In living organisms, iodine is involved in the synthesis of thyroid hormone and other organoiodides, in pregnant and lactating women it is transferred to their fetuses and newborns as a nutrient, it is an element involved in antioxidant (damage control), anti-inflammatory and antimicrobial activities (infection prevention), and it also has antiproliferative, proapoptotic and prodifferentiation properties (growth and cancer prevention) [5,6,7]. In contrast to popular belief iodine is not only located in the thyroid and incorporated in thyroid hormones. The total body iodine content is about 30–50 mg and less than 30% is present in the thyroid gland and in its hormones. Approximately 60–80% of total iodine is nonhormonal and concentrated in extrathyroidal tissues, and 23% is in the gastrosalivary pool [5,6,7,8].

The selenium atom (^80^Se) is composed of 34 protons and 46 neutrons. It is an element of group 16 (VIa), period 4, of which the outer 4p-orbit is filled with 4 electrons ([Ar] 3d^10^ 4s^2^ 4p^4^). There are 6 naturally occurring isotopes with neutron numbers 40 (^74^Se; 0.87%), 42 (^76^Se; 9.02%), 43 (^77^Se; 7.58%), 44 (^78^Se; 23.52%), 46 (^80^Se; 49.82%) and 48 (^82^Se; 9.19%; beta-emitter with very long half-life of 1.4 × 10^20^ years), giving rise to an atomic mass of 78.96. Chemically and physically selenium behaves as an intermediate between a metal and nonmetal. The oxidation/reduction states in its chemical forms are 2− (selenide, Se^2−^), 0 (elemental Se), 4+ (selenite, SeO_3_^2−^) and 6+ (selenate, SeO_4_^2−^) [9].

Living organisms contain selenium mainly as organic selenium, notably selenomethionine and selenocysteine, in which the sulfur atoms (16 protons, also group 16, but period 3) in methionine and cysteine are replaced by selenium [9]. Most of the activities of selenium in organisms are attributed to the currently known 25 selenoproteins that contain selenocysteine as the 21st proteogenic amino acid. They have many functions in their capacity as antioxidants, in redox signaling, in the metabolism of thyroid hormone and the liberation of iodide, in protein folding and in selenium transport and storage, while some of the functions are as yet unknown [9,10,11,12,13].

The high similarity of selenium with sulfur, with regard to oxidative states, explains their similarity in metabolism, although selenocysteine is biosynthesized differently from its sulfur-cysteine analog. There might be two metabolic selenium pools. The main derives from inorganic selenite and selenate and is an active pool for selenoprotein synthesis. The second is the release of selenoamino acids, in particular selenomethionine, from their protein structures during protein turnover [9,13,14]. Selenocysteine is highly reactive in redox reactions and maintained at appropriate levels, whereas the less reactive selenomethionine is metabolized as the essential sulfur analog methionine. The toxicity of selenium might in part relate to the apparent similarity of selenomethionine with methionine, causing its incorporation in place of methionine as a seemingly unregulated selenium store [15]. The C–Se bond is weaker than the C–S bond [16] and there might be subtle, but possibly relevant, conformational changes in proteins in which selenoamino acids replace their sulfur counterparts [17]. Selenomethionine is, compared with methionine, extraordinarily redox active under physiological conditions, providing protection against reactive oxygen species and other possibly harmful oxidants. Selenomethionine might be the 22nd amino acid needed in proteins with functions of its own [18]. Selenotoxicity is, however, much more complex as it has a number of interacting substances [19]. Collectively, selenium’s toxicity is usually attributed to its ‘disturbance of redox balance’ [14]. Hydrogen selenide (H_2_Se) is the most toxic selenium compound by inhalation and sodium selenite (Na_2_SeO_3_) the most toxic via ingestion [9].

## 3. World Distribution and Food Chains

Both iodine and selenium are released from underground locations into the atmosphere and seawater by volcanic activity and fissural fault emission [9,20,21,22,23,24,25]. They also become available to the environment by decomposition of dead organisms [22,26]. The iodine and selenium concentrations of seawater amount to about 60 and 4 µg/L, respectively. These levels have probably not changed over millions of years [27], apart from local selenium concentrations due to anthropogenic activity (see below). In contrast, the iodine and selenium contents in the soil exhibit a patchy pattern across the world, with some places containing high levels and others very little [28,29].

Global iodine distribution is closely associated with the water cycle [20,29]. The oceans are the largest reservoir and source of virtually all iodine on the land [26]. Iodine accumulated by algae is either emitted to the atmosphere as I_2_ gas or volatile iodocarbon compounds (CH_3_I) or buried in marine sediment which may contain 70% of the iodine present in the crust and seawater [21]. Molecular iodine and methyl iodide evaporate from the sea and rain down via atmospheric precipitation to flow back to the sea, where it is present as iodate (IO_3_^−^) and iodide (I^−^), with some molecular iodine (I_2_) and iodinated organic compounds [20,26]. This biogeochemical cycle explains why iodine deficiency of people living at the seashore is rare and that the most severe deficiencies are usually encountered in the highlands (e.g., Himalayas, Andes), rain-shadow areas and central continental regions [29,30]. The iodine content in soils range from <0.1–150 mg/kg of which the majority comes from the atmosphere and ultimately the marine environment [29]. Consequently, in near coastal areas crops and vegetables may provide sufficient iodine, but inland levels are usually low. In general, plants contain little iodine, which explains why 25% of vegetarians and 80% of vegans may be iodine deficient [29]. The iodine contents of estuaries and river sources is typically less than 0.26 µg/L (Triassic mountains of northern Italy) as compared to 50–60 µg/L in the sea, while saltwater fish (herring) may contain 520 µg iodine/kg and freshwater fish (trout) only 20 µg/kg [8]. Cow’s milk can be a good source of iodine, but its content depends on many factors, including iodine intake (feed), goitrogen intake, milk yield, season, teat dipping with iodine-containing disinfectants, type of farming and processing [31,32]. Dutch cow’s milk contains 12.4–19.9 µg/100 g and thereby covers 25–40% of the US RDA for adults at a 300 mL intake [33].

The selenium biogeochemical cycle is complex because of selenium’s wide range of oxidation states and its availability in different chemical forms (inorganic and organic) and physical forms (solid, liquid, and gas) [22]. Selenium is recycled via the atmospheric, marine and terrestrial systems. Nowadays anthropogenic activity is the major source of the selenium flux, and the marine system is the natural pathway [9]. Phytoplankton bioconcentrates selenium by a factor 100–2600 and incorporates it into amino acids and proteins. Together with soil microorganisms and plants, phytoplankton species are sources of biogenic selenium volatilization, releasing it as dimethylselenide and dimethyldiselenide [34]. As a consequence, selenium is present in varying quantities in the environment [35] and today it is derived from both natural and anthropogenic sources [9]. Well known is the selenium-poor belt in China that may at some places border to selenium-very-rich places [36,37] that are linked to Keshan disease (cardiomyopathy; selenium deficiency) [35,38] and the occurrence of selenosis (selenium toxicity), respectively.

Selenium delivery to the terrestrial food pyramid depends mainly on the levels of plant-available selenium in the soils that are used for agriculture. Soil selenium contents reach an average of 0.4 mg/kg, and range from 0.01 up to 1200 mg/kg. Selenate (Se^6+^), being the most oxidized selenium form, is best bioavailable to plants, especially under alkaline conditions. As a result, selenium in plants may range from 0.005 to 5500 mg/kg, but most contain less than 10 mg/kg [9]. That renders selenium as a critical nutrient in vegan and vegetarian diets, especially in selenium-poor areas. Plants accumulate selenium (mostly as selenomethionine) depending on the species, to levels generally in proportion to the soil contents. They do not need it as an essential element for growth but may themselves suffer toxicity at high accumulation levels [9,15,34,39]. Nonprotein selenium may also be present in plants, notably in the so-called selenium accumulators. Brazil nuts (deriving from the Amazon) are famous for their high selenium contents [40]. Cereals grown in North America contain more selenium than European crops. Finland, a selenium-poor country, has successfully increased the population selenium status by enriching plant fertilizers [41]. Selenium levels in cow’s milk depend mainly on selenium in feed, which is also reflected by seasonal availability [42,43]. The mean level of selenium in Dutch cow’s milk was 23 µg/L, which covers 12.5% of US RDA for adults when taken in a 300 mL amount [44].

The resulting huge geographical variation of dietary selenium intake [45] is important to selenium deficiency and toxicity in humans, notably because selenium exhibits probably the narrowest window of desirable intakes of all elements, ranging from deficient (about <40 µg/day) to toxic (>400 µg/day; IOM) and a 70 µg/day adequate intake (EFSA [46]) for adults. On a global scale it is estimated that, dependent on environmental circumstances, adult dietary intakes range from 3 to 7000 µg/day, with huge differences between selenium-deficient and seleniferous regions as found within China, and between South American countries (e.g., Andes-Orinico high vs. Argentina low). Selenomethionine and selenocysteine are the major ingested selenium forms from plants and animals, respectively [9]. Organic forms of selenium are better absorbed (often >90%) from the diet than inorganic forms [47]. Of the inorganic forms, selenate is absorbed almost completely, but a significant fraction is lost in urine before it can be used. Selenite, has a more variable absorption but is much better retained than selenate [48]. The total body selenium content in Western countries has been estimated at 3–20.3 mg, with muscle (27.5%), bone (15.9%), and liver (7.66%) harboring most [49]. The testes has priority when the selenium status is low [9], which may point at the high evolutionary value of reproduction. Selenoprotein P is a selenocysteine-containing protein that carries selenium from the liver to tissues that express the apolipoprotein E receptor-2; the uptake system of selenoprotein P. At low selenium status, cells utilize selenium to synthesize only the selenoproteins most important to them. The combination of uptake system and cellular selenoprotein preference creates a complex hierarchy that favors expression of the selenoproteins most needed by that particular cell [15].

Iodine and selenium in the sea may become bioconcentrated by phytoplankton (algae) and from there work their way up in the food chain. Seaweeds are not plants but macroalgae. Especially brown seaweeds may concentrate iodide up to 30,000 times [50,51,52]. Phytoplankton may also be an important source of the polyunsaturated (‘fish oil’) fatty acids eicosapentaenoic acid (EPA) and docosahexaenoic acid (DHA). Jointly, iodine, selenium, iron, zinc, copper, vitamins A and D, vitamin B_12_ and EPA/DHA have been coined ‘brain selective nutrients’ by Cunanne, Crawford and Broadhurst. Each of these is abundantly present in the so called ‘land–water ecosystem’ [53,54,55,56,57,58,59]. They are needed for normal human brain development, but are not ‘specific’ to the brain, since they are also used elsewhere in the body. However, if the requirement for any one of these is not met at the correct stage of development, permanent retardation results [57]. At least three of these, iodine, selenium and (heme) iron, are necessary for thyroid hormone synthesis [60,61,62,63,64,65,66,67].

In red algae, iodine and selenium may accumulate to high levels, in some reaching 2407–3108 µg iodine and 126–204 µg selenium/g dry weight. This implies that for adults, to reach the current EFSA intake recommendations [46,68], less than 200 mg of dry weight is sufficient to reach the 150 µg/day iodine adequate intake, whereas less than 500 mg provides the 70 µg/day adequate intake of selenium. Many mollusks filter-feed on algae and may accumulate 160 µg iodine [69,70], and 19–77 µg selenium per 100 g, depending on location [71]. Commonly available seaweed and crustaceans/mollusks in Hong Kong contain 84–290,000 (mean 46,000) and 3.2–610 (mean 97) µg iodine per 100 g, respectively [31]. The traditional Japanese diet is rich in seafoods and in 1964 may have supplied 13.8 mg iodine per day [7]. An intake of more than 20 mg iodine/day from 10–50 g seaweed has been reported at the coast of Hokkaido (the northernmost island of Japan) [7,72]. This high seaweed intake has been related to a form of endemic goiter that is clinically neither hypothyroid nor hyperthyroid, causes accumulation of inorganic iodine in the thyroid and seems at least in part reversible upon discontinued intake [72]. It has also been named ‘endemic coastal goiter’ and disappeared in the late 1970, while high urinary iodine excretion continued in the subsequent 20 years. Even today the urinary iodine concentration of school-age children in the Hokkaido main island (median 312 µg/L) and two nearby small islands (medians 633 and 699 µg/L) rank the highest in Japan with the majority of them exceeding the 300 µg/L ‘excessive’ intake of the WHO [73].

Apart from a high iodine intake, the Japanese also eat high amounts of selenium (151–191 µg/day; [74]. Their diet may contain up to 500 µg selenium/day from sea fish [9,75]. Consequently, the Japanese also have much higher selenium in skeletal muscle (1700 ng/g), compared with people from Canada (370 ng/g), W-Germany (111 ng/g), New Zealand (61 ng/g) [76], USA (400 ng/g), Scotland (170 ng/g) and Poland (51 ng/g) [49].

## 4. Evolutionary Background

About 90% of the history of life took place almost entirely under water. It took billions of years for life to conquer the land beginning about 400 million years ago [77]. This section discusses how evolution made use of iodine and selenium properties.

### 4.1. Iodide as Antioxidant

As explained by Venturi and coworkers, early oxygenic photosynthetic organisms probably already accumulated iodide (I^−^) from the surrounding seawater [3,20]. They used it to counteract oxidative stress, as experienced from dissolved oxidants, which comprise the reactive superoxide anion-radical (·O_2_^−^), hydroxyl radical (·OH), ozone (O_3_) and H_2_O_2_ that (catalyzed by Fe^2+^) may be converted to the highly reactive hydroxyl radical (Figure 1). In the reaction between iodide and such an oxidant, iodide becomes oxidized to molecular iodine (I_2_) or methyl iodide (CH_3_I). The electron released from iodide in this process is transferred to the threatening, highly reactive, oxygen radical, which contains an unpaired electron that eagerly searches for a free electron to form a ‘pair’. Accordingly, iodide (I^−^), as a ‘source of electrons’, was probably used as an early inorganic antioxidant [20]. Today this reaction can still be observed in the form of the molecular iodine and methyl iodide, both gases, that escape from the kelp fields as observed at the Southern Californian coast [50,78]. Especially when falling dry during low tide, seaweeds may experience oxygen toxicity. Also in humans, iodide appears to be an effective destroyer of reactive oxygen radicals in blood cells [50], but in humans iodide as antioxidant has not yet been studied in detail.

### 4.2. Iodine as Oxidant, the Peroxidase Partner System

In the reverse reaction, iodine (I or I_2_) is converted to iodide (I^−^), thereby gains an electron, and thus becomes reduced. The electron provider becomes oxidized and if that substance happens to be an important part of a bacterium, for example, a cell wall component or its genetic material, it may be killed. This is a mechanism of action similar to iodine tincture: a widely used disinfectant water-alcohol solution containing iodine and potassium iodide (Lugol). Using this solution, there is a strong effect of the molecular iodine (I_2_), which reacts rapidly with components of viruses, bacteria and fungi [79].

With a system similar to kelp, we kill microorganisms with the so-called peroxidase partner system: evolution continues on a successful path (Figure 1). However, the oxidative stress is not from the outside; we produce it (H_2_O_2_) ourselves at the cell–environment interface. At that location, the produced H_2_O_2_ reacts with iodide that has been transported into the cells by the Na^+^/I^−^ symporter (NIS) and subsequently excreted at the apical site. The reaction is catalyzed by peroxidases (e.g., gastric peroxidase and lactoperoxidase) and gives rise to the formation of reactive iodine species (like I_2_ and OI^−^) at the lining of exocrine glands, thus transforming iodide into an antimicrobial agent. Not surprising, the peroxidase partner system is found at places where our body borders to micro-organisms and cannot escape communication with the outside, notably the exocrine glands. These organs are vulnerable to infection, such as the salivary glands, mammary gland, prostate, pancreas and others, but also the gastrointestinal mucosa and surface of our lungs [4].

The organs using this system betray themselves by rapidly accumulating radioactive iodide injected into the bloodstream but not storing it like the thyroid does. Radioactive iodine has been shown to recycle through these organs like the salivary glands and stomach wall, to accumulate progressively in the thyroid gland, and become excreted via the urine [2,20,69,80]. Figure 2 shows that the tracer is rapidly taken up and recycled by the oral mucosa salivary glands, gastric mucosa, and excreted via the bladder. The thyroid is the organ that accumulates the tracer. This process points at the need of a constant dietary iodide source to serve iodide’s function in the peroxidase partner system located in extrathyroidal organs [81].

All of these organs have in common an iodide transporter (Na^+^/I^−^ symporter: NIS) [82,83], a H_2_O_2_ generating system (e.g., dual oxidase: DUOX), and peroxidase enzymes (the peroxidase partner system) [4]. The NIS is expressed in several tissues such as the thyroid, salivary glands, gastric mucosa, lactating mammary gland, choroid plexus, ciliary body of the eye, lacrimal gland, thymus, skin, placenta, ovary, uterus, prostate, and pancreas [82]. The H_2_O_2_ generating systems are mostly dual oxidase (DUOX 1 and 2), xanthine oxidase, or glucose oxidase [4,84,85], while the (heme-iron) peroxidases may be thyroid peroxidase (TPO; thyroid), gastric peroxidase (GPO; stomach), salivary peroxidase (SPO; salivary glands), myeloperoxidase (MPO; leukocytes) and the well-known lactoperoxidase (LPO; salivary glands, intestinal mucus, tears, milk) [4,86,87].

Catalyzed by thyroid peroxidase (TPO) reactive iodine species in thyroid colloid react with tyrosines in thyroglobulin, which constitutes the first step in thyroid hormone synthesis (Figure 1). Thyroid hormone (T_3_) is crucial for metamorphosis in amphibians (its initial use) and for (brain) growth and metabolism in humans. Thyroid hormone synthesis is a highly regulated modification of the peroxidase partner system in exocrine glands. The antioxidant properties of selenoproteins (e.g., glutathione peroxidase; GPx) are needed to prevent toxicity of any H_2_O_2_ and other reactive species that endanger the cell, while the selenium-containing iodothyronine deiodinases (DIOs) are needed for the liberation of iodide from an organoiodide store, and for the production and deactivation of triiodothyronine.

As said, the peroxidase partner system aims at the destruction of microbes, such as viruses, bacteria and fungi [4,84,86,88,89,90,91,92]. In the extrathyroidal tissues [82], the system also makes use of other substrates that are transported by the NIS, and are usually considered ‘goitrogens’, like thiocyanate (SCN^−^), bromide (Br^−^), chloride (Cl^−^; see below) and possibly selenocyanate (SeCN^−^). With the appropriate peroxidase and H_2_O_2_ these substrates become converted to hypoiodite (OI^−^), hypothiocyanite (OSCN^−^), hypobromide (OBr^−^), hypochlorite (OCl^−^) and hyposelenocyanite (OSeCN^−^), respectively. Jointly, these reactive compounds constitute a kind of ‘broad spectrum’ antibiotic that targets different microbial functional groups [89]. The optimal combination of these substrates is currently unknown, but provides a different view on the ‘evilness’ of goitrogens. Only chloroperoxidase, eosinophil peroxidase and myeloperoxidase can oxidize the highly electronegative chloride (to OCl^−^) and this reaction is used in the so-called respiratory burst employed by granulocytes; also to kill microbes [88,93,94]. For this purpose, hypochlorite is widely used in swimming pools and as toilet disinfectant. The antiviral and antibacterial activity of the lactoperoxidase system has shown substrate-specific outcomes with possibly the best effects in case of iodide and selenium [84,95]. In line with this, selenium status above that required for optimal selenoprotein function has been associated with better cure rate from COVID-19 [96]. In addition, selenium deficiency is associated with increased virulence of certain viruses through accelerated mutation [97,98], which is probably the main pathophysiological basis of Keshan disease [38,98]. The etiological origins of viral infectious disease, like HIV, some influenza viruses, Ebola and SARS-CoV-1 and 2, correlate with geographical regions of poor selenium bioavailability from the soil [35], and for 17 cities in China there is a striking association between COVID-19 recovery rate and (hair) selenium status [99,100].

### 4.3. Evolution Explains the Present

In view of the early evolutionary systems at least three subjects seem of importance to understand current (patho)physiology: (1) the early use of iodoorganic molecules for storage and transport of iodide, (2) the use of iodothyronine deiodinases (DIOs) for the metabolism of T_4_-to-T_3_ with concomitant liberation of iodide, and (3) the evolutionary more recent modification of the peroxidase partner system to produce thyroxine (T_4_; prohormone) and triiodothyronine (T_3_, hormone) by thyroidal TPO (Figure 1).

Iodine reacts easily with aromatic systems, like tyrosine and histidine, and also with polyunsaturated fatty acid double bonds [20]. This feature was probably used by early organisms to store and transport iodide for its later use as an antioxidant (I^−^ to I_2_) and probably also by the peroxidase partner system as an oxidant (I^−^ + H_2_O_2_ to I_2_ and OI^−^) for killing microbes. In other words, the mono- and di-iodotyrosines, T_4_, T_3_ and probably others might have had original functions for iodide storage and transport. That is: not as precursors of a hormone (T_3_).

Second, the primary functions of the selenium-containing iodothyronine deiodinases (DIO1 and DIO3) might have been the release of iodide from a storage molecule, and not the synthesis and degradation of a hormone. The connection between infection and the conversion of T_4_ to reverse T_3_ by DIOs 1 or 3, resulting in the liberation of iodide from an iodide store for its subsequent use in the antimicrobial peroxidase partner system, supports this thought. Accordingly, DIO2 is the phylogenetically youngest and most subtly regulated, which agrees with its important function in converting T_4_ to the active T_3_ and DIO2′s key role in neurogenesis [51]. In this reasoning it is remarkable that in most DIO metabolic schemes the liberated iodide is not shown. Cold blooded amphibians (like the frog) started to use T_3_ for metamorphosis [3], in which the signal was transduced via the phylogenetically oldest T_3_ receptor, the alpha-T_3_ receptor (600–500 million years ago), while in mammals T_3_ acquired functions in metabolism, thermogenesis and neurogenesis via the younger beta-T_3_ receptor (250–150 million years ago) [8,80,101]. Selenium was essential for functions in both the DIO’s and the degradation of excessively generated, potentially dangerous, H_2_O_2_, via glutathione peroxidases (GPXs), thioredoxin reductase (TRX) and others [87], rendering the thyroid the organ with the most selenium per gram tissue [102,103]

Finally, the thyroid is embryogenetically and phylogenetically, derived from the primitive gut [3]. Thyroid cells are, thus, evolved gastroenteric cells which, during evolution, migrated and specialized in the uptake of iodide, and in the storage and elaboration of iodine compounds. The thyroid has the typical anatomy of an exocrine gland in which the ‘excretion product’ becomes, in an unusual way, transported from the lumen to the circulation to behave as an internally-secreting endocrine gland. Iodine storage enabled the transition from an iodine-rich ocean to an iodine-deficient terrestrial environment beginning some 500–600 million years ago [3,101] with centipede-like animals exploring the world above water, followed by plants some 430 million years ago and finally the crawling of prehistoric fish out of the water some 30 million years thereafter [104].

Thyroid hormone synthesis [87] is thus a modification of the peroxidase partner system (Figure 1) that became highly regulated because of the crucial function that T_3_ eventually gained in (brain) growth and metabolism of virtually every cell. For instance, the gastric iodide pump (NIS) is phylogenetically more primitive than the thyroidal one, has lower affinity for iodide and does not respond to the evolutionary more recent thyrotrophin (TSH) [101]. This renders extrathyroidal iodine metabolism sensitive to environmental influences and the weakest in maintaining appropriate function when bodily iodine, and probably selenium status, become critical. Such hierarchies have become known as the ‘triage hypothesis of Ames’ [105,106,107] and were further detailed by Burk and Hill [15].

Taken together, the risk of iodine and selenium related disease started when life moved from the sea to land and came in a free fall in modern times in which humans changed their seashore diet for what they eat today.

## 5. Substantiation of Current Recommendations

The triage theory of Ames posits that when micronutrient availability is limited, functions required for short term survival take precedence over functions whose loss can be better tolerated. This may lead to, among others, chronic diseases in the ageing population [105]. In other words, defining a recommended intake should also take the long-term consequences into account when possible, while these are the most difficult to study. As an example, the higher-than-recommended intake of iodine by Japanese mothers (>176 µg/day) was associated with improved neurodevelopment in childhood [108]. Selenium deficiency is linked to several cancers, such as those from the prostate and breast [109,110,111]. The selenium intake associated with lowest breast cancer risk, was about four times higher than currently recommended [112]. Changes in diet, amongst other in the intake of iodine and selenium, may be an important factor to explain why prostate and breast cancer increased in Japanese USA immigrants [113,114,115,116]. Therefore, the triage theory suggest that currently recommended amounts may not always cover the total requirement of a nutrient and may thereby explain the success of multi vitamin and mineral supplements [97,105,117,118].

The determinants used for establishing iodine requirements for adults by a number of influential institutes (US Institute of Medicine, FAO/WHO, and European Food Safety Authority), as well as the derived dietary reference intakes for all other age groups, are listed in Appendix A. Dependent on age group, apparently all institutes focused on the relation between iodine intakes and thyroid function, the amounts provided by breastmilk, and the related urinary iodine excretion. The latter parameter is often used to establish the iodine status of a population [119]. Unfortunately, iodide as an antioxidant and its extrathyroidal functions were not taken into account.

Symptoms of thyroid dysfunction (elevated TSH concentrations, breakdown of thyroglobulin) were also used for establishing the iodine tolerable upper intake (UL) [120,121,122]. The elevated TSH concentrations following an acute high iodide intake are caused by inhibited iodide organification, the so-called Wolff–Chaikoff effect (see also below) [123]. The herewith established Lowest Observed Adverse Effect Level (LOAEL) for healthy adults is 1800 µg/day [120,121,122]. By using an uncertainty factor of about 1.5 [120] to 3 [121], the iodine UL for all adults has been set at 1000–1100 or 600 µg/day. In countries with long-standing iodine deficiency disorders, the intake should not exceed 500 µg/day to avoid the occurrence of hyperthyroidism [121]. ULs for children (200–600 µg/day) and adolescents (900 µg/day) were obtained by extrapolation of the adult UL [120,121]. For infants, no UL has been established [120] or are estimated to be 2–3 times the normal intakes from breastmilk [122]. A more extended description of iodine ULs is provided in the Appendix A.

For the estimation of selenium requirements in adults, the plateauing of plasma glutathione peroxidase activity (GPX3; 10–30% of plasma selenium; plateauing at 90–100 μg/L [123]) [48,121] or selenoprotein-P concentration (SEPP1; 30–60% of plasma selenium; plateauing at about 125 µg selenium/L [124]) [46] have been used (see also Appendix A). SEPP1 is considered to be a selenium storage protein and is currently the most sensitive for selenium status [46,122]. For infants, recommendations are based on levels in breastmilk. Recommendations for children are extrapolated from adult estimates (taking weight and metabolic rate into account) [46,122], or based on the prevention of Keshan disease [48]. Keshan disease, however, may depend on many other stressors [98]. The recommendations for selenium are summarized in Appendix A. Urinary selenium may rather reflect the more recent selenium intake than the nutritional selenium status [48]. For the UL, chronic selenosis with hair loss and nail brittleness as the most important parameter (most frequently reported) has been used [48]. Studies in China indicated a LOAEL for selenosis of 900–1000 µg selenium/day [121]. An intake of about 800 µg selenium/day (serum level 968 µg/L) in 5 Chinese subjects from a seleniferous area [125], and a chronic intake as high as 724 µg/day (on average 239 µg/day) in US adults, did not result in selenosis [126]. Based on these data, 800 or 850 µg/day have been used by the IOM and EFSA, respectively, to calculate the UL for all adults (400 and 300 µg selenium per day), using uncertainty factors of 2 and 3. The FAO/WHO provisionally suggested an UL for adults of 400 µg/day. For young infants, 0–6 months, a human milk with 60 µg/L that did not show known adverse effects was used to calculate the IOM UL [127]. The intake of selenium per kg body weight in adults was used to calculate the UL for older infants, children, and adolescents [48,121]. A more extended description of selenium ULs is provided in the Appendix A.

Recommended intakes of iodine and selenium should be fulfilled by the normal diet. However, as discussed, the current Western diet may not be able to cover their needs in all cases. At least for specific areas and target populations, additional iodide intake is recommended either by fortification of a general food or by taking supplements. In many countries iodide is added to salt or to a variety of foodstuffs [128]. In this manner bread became the principal source of iodine in many countries. Pregnant and lactating women are at highest risk of micronutrient deficiencies, including those of iodine and selenium. At present, multiple medical societies recommend iodine supplementation for women who are pregnant, lactating, or planning pregnancy [129]. Selenium supplementation is not recommended in pregnancy, but according to the American Thyroid Association ‘account has to be taken of different intakes of iodine, selenium or both, in different regions before any action is undertaken regarding selenium supplementation’ [130].

## 6. Considerations Pertaining to Iodine and Selenium Dietary Reference Intakes

As described earlier, there are many variables that need consideration to arrive at scientifically based recommendations for the optimal intake of any nutrient. Some investigators [131] believe that for determining iodine and selenium recommendations, the whole of evidence should be used as originally defined for Evidence Based Medicine (EBM) by David Sackett [132]. Only evaluating randomized controlled trials (RCTs) or meta-analyses, is a limitation not supported by the original definition. In the following we discuss a number of issues that are usually not taken into consideration when establishing dietary reference intakes.

### 6.1. Iodine and Selenium Regulation

Our bodily iodine homeostasis is not well regulated to counteract an impending shortage. Iodine may be stored in the thyroid for the local, highly important, thyroid hormone function, but it is also clear that iodine intake may be estimated from iodine excretion in urine [133]. The renal clearance of iodide is about 30–50 mL/min, meaning that iodide is poorly retained. Ninety percent of dietary iodine is excreted in urine in the subsequent 24–48 h [134]. Poor regulation may also be concluded from the rapid dose-dependent increases and decreases of iodine in human milk [135], the association between milk iodine and iodine status in various countries, and the fact that infants are born with limited reserves [136]. The rapid circulation of iodine through the exocrine glands [69] suggests that there is notably poor retention of iodine for use in the peroxidase partner system and that a constant iodine supply is needed to fulfill extrathyroidal needs.

Selenium homeostasis is more difficult to comprehend because of the many selenium forms and its excretion via urine, feces and the lungs. Incorporation of ingested selenomethionine in tissue proteins, notably muscle, may build a store that is nevertheless slowly released according to the local protein turnover rate. Renal selenium clearance has been estimated at 0.18 mL/min [137] which is consistent with the fact that most is incorporated into the circulating (selenium-rich) selenoprotein P. About 50–60% of ingested selenium is excreted via urine, a parameter that can be used to estimate selenium intake [138]. Also milk selenium varies with selenium in the local soil [139]. In lactating women both plasma selenomethionine and selenite respond to their supplementation, but only supplemented selenomethionine responds in milk [140]. Like iodine, infants are born with limited selenium reserves [136]. Schomburg suggested that subjects with suboptimal selenium status who are confronted with a trigger, such as pregnancy, acute infection or trauma, might lower their selenium status below a certain threshold that increases the risk of a poor outcome or autoimmune disease [47].

Taken together, iodine is very poorly retained which notably renders the extrathyroidal iodine function at risk. Selenium is much better retained but might at low basal status not be readily available from its protein store when the need is suddenly increased. As a rule, what has not been regulated in evolution did not require regulation, suggesting that there was limited evolutionary pressure on iodine and selenium. Both elements come together in our ancient diet that was abundantly available from the sea. What has not been regulated also makes one vulnerable upon environmental change (such as a changing diet). According to Darwin, there was first an environment and then came an adapted form of life. If environment (nurture) changes, physiology (nature) adapts by selection, with disease and death for those who don’t. This principle has even become the new definition or health [141] and has become popular by the expression ‘adapt or perish’.

### 6.2. Our Diet Has Changed

Deriving from the seashore ecosystem, and in view of the patchy distribution of both elements among terrestrial soils and foods, and also the variable selenium (bio)availability [19], it looks unsurprising that iodine and selenium deficiency and selenium toxicity were among the major challenges that life encountered since it conquered the land from some 400 million years ago. From isotope studies it became clear that the combined agricultural and animal domestication revolution [142], beginning some 10,000 years ago in the Middle East, i.e., a spilt second ago in evolutionary terms, has initiated a steady decline in the consumption of seafood [143,144]. Also recently in Japan, where the consumption of fish and shellfish decreased linearly from a steady 95 g/day until 1997, to 73 g/day in 2011 [145].

The National Health and Nutrition Examination Surveys (NHANES) in the USA showed a 45% decrease in urinary iodine levels from a median of 320 μg/L in 1988–1994 to 145 μg/dL in 1971–1974 [146]. In 2008, 60% of iodine in The Netherlands came from iodinated salt [147]. The lowering of iodine in baker salt from 70–85 to 50–65 mg/kg in 2008 has decreased the iodine intake in The Netherlands by 33% [148]. The genuine drop by this action may have been counteracted in part by higher iodide in milk, at least as observed in the Netherlands [149]. At present many pregnant Dutch women have low iodine status [150]. They also have poor selenium status and 61% are both iodine and selenium deficient [151]. Currently 88% of the global population uses iodized salt, and the number of countries with according to the World Health Organization (WHO) adequate iodine intake has nearly doubled from 67 in 2003 to 118 in 2020. However, still 21 countries remain deficient whereas 13 countries have excessive iodine intakes. It is important to realize that much information is based on schoolchildren as proxy for the general population and that 100–299 µg/L urinary iodine was used for ‘iodine adequacy’ and >300 µg/L for ‘excessive’ [152].

The above data are disturbing, since even mild maternal iodine deficiency may adversely affect child neurodevelopment [129]. For instance, inadequate iodine status of pregnant women in the UK and Tasmania relates to lower IQ and educational outcomes of the offspring at 8–9 years [153,154,155], while iodine supplementation improved perceptual reasoning in mildly iodine-deficient 10–13 years old New Zealand children [156]. According to the WHO, iodine deficiency is the single most important cause of preventable brain damage globally. Children born in iodine-deficient regions have on an average 13.5 intelligence quotient (IQ) points less than children born in iodine sufficient regions [157].

### 6.3. Ancient Diet and Brain Selective Nutrients

The genuine answer to our optimal iodine and selenium intakes, and also of EPA and DHA, and as a matter of fact all nutrients, comes from the food that our ancestors have eaten for millions of years [54,58,158,159,160]. That ‘optimal’ diet, as part of ‘environment’ (nurture), is the one that has driven our current physiology (nature) and is increasingly considered to be a diet that was available to us at the seashore. The still dominant but incorrect hypothesis, of our derivation from the savannah is unlikely for many reasons, e.g., because of the lack of local fossils, the daily needed large amounts of water and sodium to prevent dehydration in a hot climate, and the lack of advantage to hunt in an open plain with a relatively slow moving body in an upright position [161,162,163,164,165]. But also because of the scarcity of iodine, both the scarcity and abundance of selenium in many places on the land, and our poor ability to synthesize the fish oil fatty acids EPA and DHA from their precursor essential fatty acid alpha-linolenic acid that is abundantly present in terrestrial plants [164,166,167,168].

The East African Rift Valley has without doubt been a cradle of humans because of its rich archeological record [169,170,171,172]. In that inland region, both iodine and selenium may come from the locally (still) high volcanic activity [173] and the many alkaline salt lakes where the water flows in and subsequently evaporates, such as Lakes Turkana-Kenya and Natron-Tanzania. Well known are the footprints preserved in wet volcanic ash at the Laetoli excavation site (Olduvai gorge, Rift Valley, Tanzania), probably left behind by members of the species Australopithecus afarensis some 3.66 million years ago [174]. Worldwide archeological records show the consumption of lake, river and sea animals (e.g., shellfish) [175], since homo erectus appeared (about 2.5 million years ago) [175,176,177,178,179,180,181,182,183]). Even today 70% of the world population lives closer than 5 km to water [184]. River deltas are ideal sites for human habitation because of their fertile floodplains, easy access to the ocean and abundant land [185].

A plausible background for our connection with a seashore diet comes from Marean and coworkers [176,186,187] who posit that humans went through a population bottleneck after the eruption of the Toba supervolcano in Sumatra some 74,000 years ago. Homo sapiens may have been able to survive the following decade-long volcanic winter by the consumption of food from the nutrient-rich South African seashore, such as shellfish, to repopulate the entire world since then. In such places, shellfish and seaweed are abundantly present and easy to harvest at the seashore [59]. Coastal resources are less susceptible to such an eruption than the plants and animals of inland areas [176,186]. The presumed derivation from South Africa is in line with studies indicating the locally highest genetic variation (highest fixation index and lowest linkage disequilibrium) [188], while there is loss of heterozygosity (cline) moving to North Africa and from there to the rest of the world [189]. It has also become clear that the Out-of-Africa diaspora occurred along the seashores, which in those days stretched much farther from the present coastline because of the reigning ice age [190,191,192].

Shellfish are an abundant source of many nutrients that nowadays seem difficult to obtain. Not only iodine and selenium and fish oil fatty acids EPA and DHA, but also iron, zinc and vitamin B_12_. Of these, iron [193,194], iodine [82,153,195], zinc [196] and selenium [197] are acknowledged as the world’s most prevalent micronutrient deficiencies today, while also low vitamin B_12_ status constitutes a worldwide problem [198].

Seaweed seems nevertheless to be an unpredictable source of abundant iodine which may raise concern because of its feared toxicity [199]. As noted above, the current upper limit for iodine has been set at 1100 µg/day by the IOM and is based on the occurrence of the Wolff–Chaikoff effect using an uncertainty level of 1.5 [200]. This reaction follows the sudden ingestion of a large iodine dose, causing the thyroid to shut down thyroid hormone synthesis and leading to temporal hypothyroidism, as shown by an increase of TSH, which is the primary regulator of the NIS in the thyroid, apart from iodide itself [82,103,201]. Normal adaptation to this acute Wolff–Chaikoff effect entails adjusted thyroidal NIS expression and thereby thyroid hormone synthesis [202], implying that notably the extrathyroidal NIS-carrying organs eventually benefit from a chronically increased iodide intake.

However, mildly to moderately iodine deficient 73 years old people in New Zealand with suboptimal selenium status (based on glutathione peroxidase activity) received >50 mg iodate/day (i.e., >45 times IOM UL) for 8 weeks because of a supplement formulation error. The supplement was taken with or without 100 µg selenium/day. Ten of 43 participants exposed to this excess iodate showed elevated TSH (hypothyroidism) at 8 weeks, but in all but two, TSH had returned to normal at 12 weeks. In three participants, TSH decreased to hyperthyroid levels at 8 weeks and remained low at 12 weeks. It was also found that the high iodine supplement blunted the increase of plasma selenium and whole blood glutathione peroxidase caused by the cosupplemented selenium. This probably indicates that the increased oxidant stress from the high iodate load resulted in higher consumption of glutathione peroxidase. The authors concluded that coadministration of selenium is important for the correction of iodine deficiency [203]. De la Vieja and Santisteban [4] state that adequate selenium homeostasis may contribute both to selenoprotein expression and activity, and to NIS expression and its recovery following the Wolff–Chaikoff effect. It takes time to synthesize new selenoproteins, while a sudden high iodide intake bombards the obviously defenseless NIS almost instantaneously. Indeed, it took 4 weeks to reach maximum glutathione peroxidase activity in platelets (half-life 5 days) after supplementation with selenite or selenate [204]. Taken together, the above information suggest that the NIS has not been evolved to withstand an acute high iodide intake, but that the oxidative stress from a sudden high cellular iodide influx should become neutralized by an existing adequate activity of selenoenzymes and, undoubtedly, other antioxidant systems.

To our knowledge, there are no well-controlled data on a sudden high iodine exposure of subjects with well-established adequate selenium status at baseline. However, a transient Wolff–Chaikoff reaction was also observed in 23–65 years old subjects (n = 30) in Greenland after a single sushi-with-seaweed-salad meal in which the urinary iodide/creatinine ratio increased by 385%. The serum TSH rise was brief, and the single sushi-meal-with-seaweed salad did not cause any adverse events [205]. No data were given on selenium, but Greenland is known for its high selenium intake from seafood [45]. Such ‘dangers’ of eating a Japanese meal are to our knowledge not communicated to the general public and that might indeed not be necessary when selenium status is adequate. Finally, 64% of 44 seaweed consumers (46.1 ± 12.4 years; median/range urinary iodine concentration 1200/80–14,000 µg/L) in Norway exceeded the IOM UL of 1100 µg/day, but only one had a TSH above the reference range cutoff (i.e., 4.2 vs. 4.0 IU/L) [206].

Nevertheless, some adults fail to escape from the usually transient Wolff–Chaikoff effect and remain hypothyroidic (iodide myxedema) or progress to hyperthyroidism (Jod–Basedow effect) [207]. They obviously lack appropriate thyroid autoregulation [195], and likely have thyroid autoimmunity, often unrecognized or subclinical, or another underlying thyroid disease [6,103,202,207]. Basing optimal intakes on them closes a perfect vicious circle, because many of those may have contracted this condition *because* of an iodine/selenium disbalance in the past. Moreover, dietary reference intakes are made for the healthy population and it is questionable whether those who fail to escape from the Wolff–Chaikoff effect are healthy according to the proposed new definition of health: ‘the ability to adapt and to self-manage’ [141].

### 6.4. Iodine Intake from Seafood Is Unconstrained, but Selenium Intake Is Constrained

It is clear that the traditional Japanese diet does not fulfill the criteria that were developed by the authorities who have set the current iodine upper limits. It is in this context important to re-emphasize that the traditional Japanese diet does not only contain much iodine, but is also high in selenium, although not as liberal as iodine. Selenium intake from a seafood diet seems much more constrained as compared to iodine, the latter reaching estimated intakes of 13.7 mg/day [51,52]. Selenium may be nontoxic to terrestrial plants happily growing in selenium-rich soils and thereby, upon consumption, become toxic to us. In contrast, at the constant selenium concentration of uncontaminated seawater, selenium does not seem to accumulate in an unregulated fashion in algae, which happen to be at the bottom of the aquatic food chain, and from there on find their way to shellfish, fish and others. An 8 g dry-weight portion of 14 edible European green, red and brown seaweeds provided up to 17% of the selenium RDA, but up to 32.3% of the iodine RDA [208]. Algae are considered very resistant to selenium, possibly due in part to the use of selenium in metabolic processes and the ability to sequester or excrete selenium if necessary [209]. In other words, with the constant selenium concentration of seawater they seem much more regulated in this sense than some selenium-accumulating terrestrial plants. Selenium toxicity in algae occurs at much higher selenium concentrations than those encountered in natural or impacted aquatic ecosystems [209], suggesting that the constant selenium content of seawater over millions of years has caused relatively constant selenium levels in organisms at the bottom of the food chain. Accordingly, this form of ‘selenium regulation by seawater’ at the basis of the food web may work its way up to relatively constant levels in the animals that prey on them, with probably only the old predators at the very top reaching levels that upon consumption may potentially become toxic for us. In a survey of 1100 foods in the 2008 U.S. Department of Agriculture National Nutrient Database, seafoods comprised 17 of the top 25 dietary sources of selenium [210]. There was no necessity for strict regulation of selenium in animals who preyed on these, but poor regulation also implies vulnerability upon change.

Taken together we suggest that the maximal intake of iodine from seafood is rather unconstrained, but that of selenium is constrained. It might in the past have been, and probably still is, difficult to exceed the current selenium upper limit by eating food from an uncontaminated seashore. Indeed, the selenium intake in Japan did not change from 1957–1989. The mean ± SD was 129 ± 32 µg/day. In 1977, 57% of selenium came from the consumption of 89 g fish and shellfish per day and the seaweed intake was 5 g/day [211]. The highest selenium content in seafood in Japan was encountered in alfonsino white muscle (1.27 mg/kg tissue) [212], implying that an implausible daily intake of about 300 g covers the current 400 µg selenium upper limit of the IOM, which also contains an uncertainty factor of 2–3. In addition, alfonsino muscle also contains the highest mercury levels (1.19 mg/kg), which combination reduces the potential toxicity of both elements [210]. However, it makes one wonder about the environmental conditions of these deep-sea coral fish with a life span of up to 23 years, obtained from a ‘local Tokyo market’ [212]. The nearby Tokyo Bay is indeed highly polluted with mercury [213]. The usual selenium levels of fish muscle in Japan is much lower, ranging from 0.12 to 0.77 mg/kg tissue [212]. These figures are in line with a study in Spain, showing highest selenium levels in tuna (0.567 mg/kg wet weight), swordfish (0.494 mg/kg), scad (0.346 mg/kg), sardine (0.279 mg/kg) and mackerel (0.224 mg/kg), with most (shell)fishes containing less than 0.200 mg selenium/kg [214].

The USDA Tables indicate that the highest selenium content in seafood analyzed by them is from Mollusks (oyster, Pacific, cooked, moist heat; USDA 15231) harboring 154 µg selenium/100 g [215]. It implies that more than 260 g should be consumed to exceed the 400 µg selenium upper limit. Brazil nuts (USDA 12078) may on the other hand contain 1917 µg selenium/100 g, corresponding with 400 µg selenium per 21 g, equaling the consumption of 4 nuts [216]. That is, if the Brazil nuts come from the Amazon, because their contents may vary about three orders of magnitude across several regions of South America (0.2–512 mg/kg). A single Brazil nut could provide either 11% (Mato Grosso state, Brazil) or up to 288% (Amazonas state, Brazil) of the daily 70 µg Se requirement for an adult man [40].

Also low intakes of the widely eaten seaweeds of the Porphyra genus (laver) are sufficient to reach the 300–400 µg/day selenium upper limit some. Hwang et al. [217] published exceptionally high selenium contents of 204 and 126 µg/g dry weight in the red macroalgae Porphyra tenera and Porphyra haitanenis, respectively, ‘purchased at Korean and Chinese markets’. These figures are much higher than the 0.106–0.219 [218], <0.1–1.3 [208], 2.62–11.5 [219] and 4.9–36.85 [220] µg selenium/g dry weight, as reported by others. The origin of the seaweed studied by Hwang et al. [217] is poorly defined, but it is known that seaweed selenium is dependent on species and, when cultured, the selenium concentration of the medium [221]. Another variable is the seawater selenium concentration in polluted areas in which case selenium mostly comes with other pollutants, notably heavy metals like mercury, lead, cadmium, and arsenic. With the figures of Hwang et al. [217] it may be estimated that 2 and 3 g dry weight of Porphyra tenera and Porphyra haitanenis, respectively, will provide the 400 µg selenium upper limit. Such figures may, however, not be judged in isolation. The concomitant arsenic and cadmium levels were 32–44 and 1.6–3.4 µg/g dry weight, respectively. For arsenic these are at the upper limit of the range usually reported for red macroalgae (range 1–50 µg/g dry weight; [208]). It suggests that pollution may be at stake and that, ironically, the high arsenic may function as partial ‘selenium antidote’. These data nevertheless show that, analogous to the advice to eat fish, information on seaweed contents is needed and has to become communicated to consumers.

The current selenium upper limit is based on selenosis caused by chronic high selenium intakes by Chinese and USA people eating a selenium-unregulated terrestrial diet composed of organisms that themselves obviously survived selenium-rich soils (Table 1). This was not the case when homo sapiens ate a seashore diet in the past. As said, contamination of food chains with selenium and heavy metals via anthropogenic activities may nowadays complicate affairs, notably in the more closed fresh water ecosystems, such as lakes or at locations where contaminated rivers meet the open sea [222,223]. It is nevertheless clear that selenium toxicity (selenosis) is to be taken seriously. The PRECISE study, using a 5-years 300 µg/day selenium supplement as selenium-enriched yeast, showed higher mortality after 15 years, compared to a placebo yeast in Denmark, which is a country with moderately-low selenium status [224]. It should, however, also be kept in mind that, in genuine food, selenium does not come on its own, and that there are many potentiating and ameliorating factors involved such as: polymorphisms, exposure to interacting toxic elements (As, Pb, Cd, Hg), speciation (selenomethionine, inorganic forms) and the microbiome [19].

### 6.5. Iodine and Selenium Interact in Thyroid Function

Combined selenium and iodine deficiency in early life is implicated in the pathogenesis of both myxoedematous cretinins (mental retardation, poor growth) [9,225,226,227] and Kashin–Beck disease (endemic osteoartropathy) [9,36,197,228]. Observational studies of iodine intake and thyroid autoimmune disease reveal a U-shaped curve, indicating that both low and high intakes relate to thyroid autoimmune disease risk [229]. Figure 3 shows this U-shaped relation between urinary iodine concentration (a parameter of population iodine status) and risk of thyroid autoimmune disease in adult men and women, as compiled from a dose–response meta-analysis of 17 studies [229]. Note that the lowest risk is at a urinary concentration of about 300 µg/L, which is equal to the 300 µg/L WHO cutoff value for ‘iodine excess’ [133]. The higher risk at low status might be caused by augmented thyroidal H_2_O_2_ production stimulated by the augmented TSH of iodine deficiency. High H_2_O_2_ production needs optimal selenium status to prevent damage from reactive oxygen species. The increased risk at high iodine status may derive from augmented production of reactive iodine species that become insufficiently detoxified at suboptimal antioxidant status provided by the selenoprotein antioxidants. Also the relation between iodine status and the prevalence of subclinical hypothyroidism among 7190 pregnant women is U-shaped [230].

Although selenium intake versus the incidence of various diseases shows a U-shaped curve as well, only low selenium intake is related to thyroid autoimmune disease. At low concentrations, immune function goes down, viral virulence goes up, and prostate cancer increases. An adequate selenium status decreases cancer risk, type 2 diabetes, supports cognition and decreases viral virulence. At high intakes: selenosis may develop, and cancer risk increases as well as type-2 diabetes [45].

The combination of the iodine and selenium U-shaped curves prompted Rayman [65] to conclude that regions with: (1) deficient iodine intakes, (2) more-than-adequate iodine intakes and (3) high iodine intakes, may need more selenium owing to the capacity of selenoproteins to protect the thyroid from excessive H_2_O_2_ and from inflammation. In some areas in China the goiter incidence is *negatively* correlated with the locally high iodine content of the drinking water. In others with high iodine drinking water, iodine intake is *positively* related to what is named ‘high iodine goiter’ [29]. Interestingly the later areas seem to overlap with the Chinese selenium-poor soils [28].

The above observation is well explained by the mechanisms that are probably involved. A deficient iodine intake impairs thyroid hormone production, which stimulates the hypothalamus to release TRH, while TRH stimulates the pituitary to release TSH. TSH stimulates thyroid growth (causing Goiter) and also each step in thyroid hormone synthesis, that is: iodine trapping via NIS, iodine oxidation via TPO, thyroglobulin synthesis, iodotyrosine coupling [195] and also H_2_O_2_ synthesis by DUOX2 [87]. H_2_O_2_ in the thyroid is made in excess anyhow, when compared with the amounts of iodine incorporated into proteins, owing to the relatively high Michaelis–Menten constant of TPO for H_2_O_2_, meaning that TPO needs a high H_2_O_2_ concentration to reach half its v_max_ for the coupling of iodide to thyroglobulin. H_2_O_2_ production increases even more at high TSH, e.g., caused by iodine deficiency [231,232]. Selenium-containing enzymes (GPx, PH-GPx) participate in the protection of thyroid cells against any escaping H_2_O_2_ that diffuses into the cell and probably also to iodine reactive species made in the follicle lumen, meaning that optimal protection by selenoproteins is even more needed at low iodine status. Thus, iodine deficiency increases thyroid H_2_O_2_ generation, whereas selenium deficiency decreases H_2_O_2_ disposal, which may in concert increase the risk of free radical damage, autoimmune disease, apoptosis and cancer. A similar condition may occur at high iodine status and low selenium status, when the surplus H_2_O_2_ probably generates abundant levels of damaging reactive iodine species, albeit in this case at normal TSH. It has been shown that iodine deficiency induces thyroid autoimmune reactivity in Wistar rats [233], that high iodine may create iodinated neoantigenic determinants in thyroglobulin to which immune tolerance has not been established [234], that the degree of iodination of thyroglobulin has major impact on its immunological properties [235], and that excess iodine may increase adhesion molecules on thyrocytes [236].

In line with the mechanisms, it was shown in an RCT that a 200 µg/day selenomethionine supplement protects against postpartum autoimmune thyroid disease in pregnant women with TPO autoantibodies [65,237]. The SETI study showed that a short-course supplementation of 50 Italian patients with subclinical hypothyroidism due to Hashimoto’s thyroiditis with 83 µg selenomethionine/day for 4 months was associated with normalization of serum TSH, which was maintained 6 months after selenium withdrawal in 50% of patients [238]. From these and other studies, Schomburg concluded that the possible side effects of selenium supplements are exaggerated, and that selenium deficiency is a risk factor for autoimmune disease [47].

### 6.6. Iodine and Selenium Interact in Exocrine Glands

Conceivably, the similarity of the reactions taking place via the peroxidase partner system in exocrine glands, may render these glands vulnerable to local damage that increases the chance of cancerous transformation and other diseases. Relatively little is known on the (lack of) local regulation of H_2_O_2_ production, and of iodine and selenium metabolism at these locations, but the NIS in extrathyroidal tissues is not regulated by TSH. Iodine tracer studies (Figure 2) suggest that these extra thyroidal locations experience a high flux of uptake and subsequent release of the iodine in some form into the lumen of the gland or directly into the GI or lungs [54,69]. Accordingly, a disbalance between iodine and selenium may explain the relation of low iodine status with cancer of the stomach, breast and prostate, and also the for long known relation between areas with Goiter and the occurrence of cancer of the stomach, but also of the breast, prostate, endometrium, ovaries, colon–rectum, and thyroid [5]. In the first USA National Health and Nutrition Examination Survey, prostate cancer risk was highest in those with low iodine status. Chronic unresolving infection because of low activity of the peroxidase partner system and/or low protection against locally produced H_2_O_2_ might be the connecting factors. Infection by microorganism, not only viruses, is an increasingly recognized cause of cancer initiation and progression [239,240,241]. The Helicobacter pylori–gastric cancer connection is just one of those [242], but microbial infection–prostate cancer is another [243,244]. By such connecting factors, having thyroid disease might also be the reason of concomitantly increased prostate cancer risk [245]. It is also known for long that the Japanese and Asians in general have remarkably low prostate and breast cancer incidence [246,247,248], while the incidence in USA-born Japanese immigrants is much higher [113]. Breast cancer incidence has increased in many countries, Japan included, but Japan has still lower incidence [246]. The mortality rate of female breast cancer in Japan and Korea has increased and is now nearing the rates observed in non-Asian countries, which in contrast have shown a steady mortality decrease [249].

In a prospective study in Korea, it was recently shown that women with breast cancer have a higher chance of contracting thyroid cancer, and that, vice versa, women with thyroid cancer have higher chance of also getting breast cancer [250]. Another Korean study showed that relatively low and extremely excessive iodine intakes are associated with thyroid cancer in an altogether iodine-replete area [251]. Iodine deficiency is associated with fibrocystic breast disease which affects many women of child-bearing age. It can be effectively prevented and treated with iodine supplementation [146]. Interestingly, an oral supplement of molecular iodine (I_2_; 5 mg/day, alone and in combination with the neoadjuvant therapy 5-fluorouracil/epirubicin/cyclophosphamide or taxotere/epirubicin) (FEC/TE) in women with early (stage II) and advanced (stage III) breast cancer improved both disease-free survival and overall survival [252].

Epidemiologically, there is an inverse relation between dietary selenium intake and mortality of cancers of the large intestine, rectum, prostate, breast, ovary, lung and with leukemia, while weak inverse associations were found for cancers of pancreas, skin and bladder, and female breast cancer mortality [112]. A recent 20 year prospective Swedish study showed that among women with high selenium levels, high iodine levels were associated with a 25% lower breast cancer risk, compared with women with low iodine levels [253]. Breast cancer mortality in the USA (on average low iodine and high selenium) and Europe (low iodine and low selenium) is about 4–5 times higher than in Japan where both selenium and iodine are high [253]. A study in China found that longevity was highest in the coastal and southern regions, having higher humidity, low standard deviation of monthly temperature, higher levels of selenium in the soil, and greater sea fish consumption. Because of high seafood consumption their iodine and selenium intakes were probably high [254]. Finally, a meta-analysis of observational studies showed that high serum/plasma and toe nail selenium have protective effects on risk of breast cancer, lung cancer, esophageal cancer, gastric cancer, and prostate cancer, but not colorectal cancer, bladder cancer and skin cancer [255].

## 7. Towards Optimal Intakes and Upper Limits

Current dietary reference intakes are based on the study of single-nutrients that are usually linked to single-disease endpoints. The outcome ignores that the human diet is composed of a balanced nutrient composition that is characteristic for the afore-living animals and plants that we eat. They also tend to be based in particular on RCTs that are widely seen as the ‘best available evidence’. However, for the development of appropriate dietary reference intakes the ecosystem in which these animals and plants live has to be understood and certainly the ecosystem that made us what we are. We argue that such an approach is consistent with the scope of ‘the whole of the evidence’ and thereby with the genuine basis of Evidence Based Medicine and Nutrition. In the case of iodine and selenium it may release us from the painful discrepancy that the current science-based iodine upper limits have been chronically exceeded by the Japanese for centuries with merely favorable effects.

It has become clear that homo sapiens has lived close to the seashore. Our evolutionary background and the many (patho)physiological studies discussed above, show that iodine intake by adults may without problems be what is currently regarded as ‘extremely high’, but that at the same time selenium intake is to be restricted. The tolerability of a higher iodine intake depends on adequate selenium status, meaning that recommendations for both elements should be linked (Table 1). Based on the traditional Japanese diet and observations in Western countries we suggest that both optimal iodine intake and its upper limit might be in the mg amounts. Based on an optimal selenoprotein P at 125 µg/L, the optimal selenium intake might be about 105 µg per day [256], which is consistent with the 129 ± 32 µg/day selenium intake in traditional Japan [211]. An intake of 105 µg selenium/day corresponds with a serum selenium of 125 µg/L and a toenail selenium of 0.74 µg/g [45]. The selenium upper limit might be kept at 300–400 µg/day. The need of a broad safety margin became strengthened by the PRECISE study [224], but a connection with ‘protecting’ toxic heavy metals should be appreciated as well. Such revised new recommendations for iodine and selenium may offer long-term benefits and better prevention of diseases that do not seem linked at first sight.

The present discussion is merely on iodine and selenium and consequently constraints to these two nutrients only. Eating from the seashore ecosystem, however, is more than just these elements. The health effects of the concomitantly consumed other brain selective nutrients, such as DHA, vitamin B_12_, iron, zinc and others, have been amply demonstrated by many investigators. Therefore, it is not a coincidence that all of these nutrients exhibit the most severe deficiencies in a world where our evolutionary origin at the seashore is not enough taken into account. From both evolutionary and (patho)physiological points of view, not sustainability, the best recommendation might be to eat daily 89 g wild fish and shellfish/day and 5 g seaweed, like in traditional Japan [211]. Terrestrial foods, such as dairy and grains, may substantially contribute to our iodine and selenium status as well, but only when adequate amounts of these elements are present in the (enriched) feeds or soil. Iodine and selenium supplements are other options but harbor the fear that disbalances with other nutrients are introduced. Whole food from a healthy environment remains best.

## Figures and Tables

**Figure 1 nutrients-14-03886-f001:**
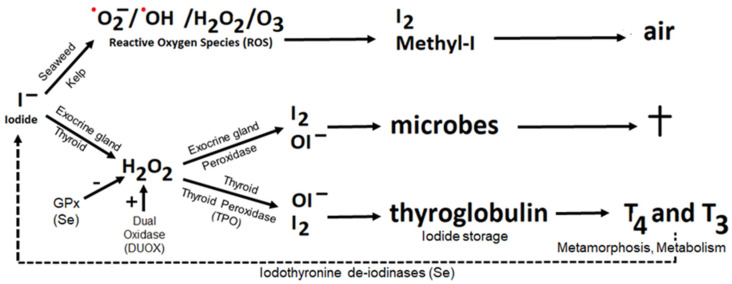
Fate of iodide as antioxidant and oxidant. *Abbreviations*. ^•^O_2_^−^: superoxide anion; ^•^OH: hydroxy-radical; H_2_O_2_: hydrogen peroxide; O_3_: ozone; I_2_: molecular iodine; methyl-I: methyl iodide; OI^−^: hypoiodite ion; GPx: glutathione peroxidase, T_4_: thyroxine, T_3_: thyroid hormone.

**Figure 2 nutrients-14-03886-f002:**
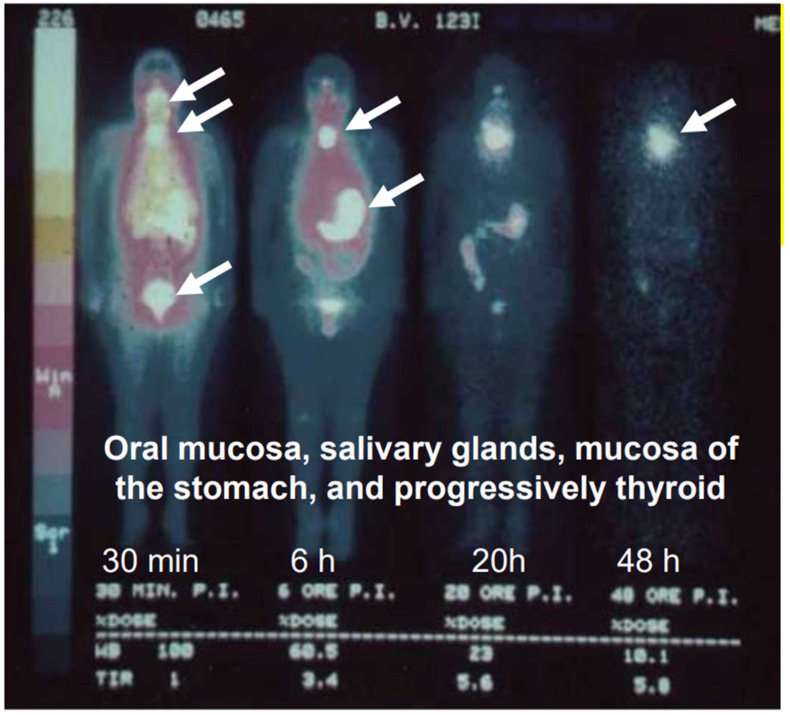
Time course of a radioactive iodide tracer following its intravenous injection. Scans were taken at 30 min and at 6, 20 and 48 h. Major trace allocation is indicated by arrows. The slightly modified figure is adapted from Venturi and Begin [20] with permission of the publisher (Wiley and Sons).

**Figure 3 nutrients-14-03886-f003:**
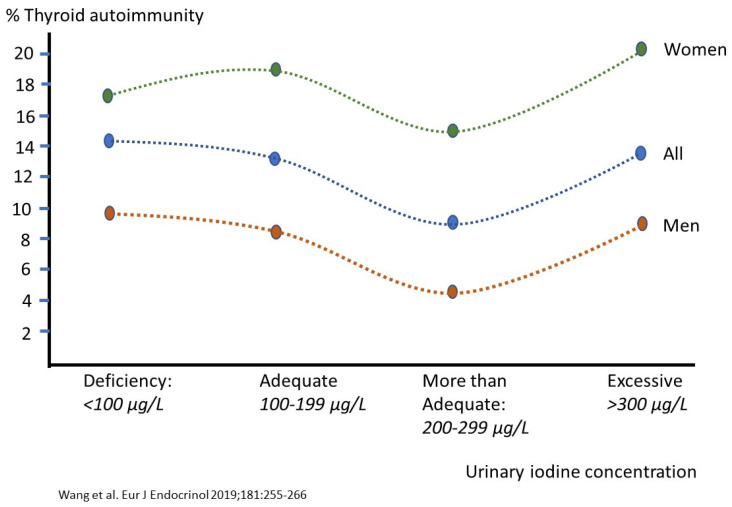
U-shaped dose-response relations for iodine and thyroid autoimmune disease in adults (n = 2802; about 50% men). Dose response meta-analysis [229]. Literature search up to 30 November 2018. Based on odds rations, ‘more than adequate’ was significantly different from Deficiency, Adequate and Excessive for ‘men’ and ‘all’, not for women.

**Table 1 nutrients-14-03886-t001:** Currently iodine and selenium recommended intakes (RDA, AI) and upper limits (UL) for apparently healthy 18–65 years old adults, their background, important information not as yet taken into account, and suggested future recommendations.

Nutrient	Current RDA or (AI) *	Basis of Current RDA or AI	Basis of Current UL	Important Information Not as yet Taken into Account	Suggested Future RDA or AI	Suggested Future UL
Iodine (µg)	150 UL: 600–1100	Thyroid iodine (^131^I) accumulation and turnover in 292 euthyroid adults, normal urinary iodine excretion (WHO: 100–199 µg/L), TSH, serum T4	Acute Wolff-Chaikoff effect causing mostly transient hypothyroidism (TSH increases). LOAEL = 1800. Uncertainty factor: 1.5–3	Selenium status Iodine’s protective effect in long-term observational studies on thyroid autoimmunity, cancer and chronic diseases mg/day intakes from the traditional Japanese/Asian diet Association with iron Associations with ‘goitrogens’ that exhibit favorable effects as anti-microbials in the peroxidase partner system	mg amounts or a very safe but conservative 300 µg, as based on U-shaped relation with thyroid autoimmunity (see Figure 3) and in view of the relatively high UL. Applies only for selenium replete subjects Protection from stomach, female breast, prostate and other cancers	mg amounts, as based on the traditional Japanese/Asian diet. Applies only for selenium replete subjects
Selenium (µg)	26–70 UL: 300–400	Plateauing of plasma glutathione peroxidase-3 or plasma SEPP1.	Selenosis. Chronic high intakes by Chinese and US adults. LOAEL= 900–1000. Uncertainty factor of 2–3, using 800–850 in the calculation	Selenium form: selenomethionine may cause delayed toxicity due to its accumulation in body proteins. Selenate is better absorbed than selenite, but less retained. Protective effect of selenium in microbial infections, prevention of virus mutation, autoimmune disease, and inflammation	105 µg, based on optimal SEPP1 at 125 ug/L (see text) May only be favorable at adequate iodine status Anti-microbial, anti-cancer and anti-thyroid autoimmunity Relation with toxic heavy metals (As, Pb, Cd, Hg) that ironically increase the selenium UL. Narrow window between RDA/AI and UL	300–400 µg, as based on the PRECISE study (see text)

RDA: recommended dietary allowance. AI: adequate intake. UL: upper limit * range of recommended intakes as derived from FAO/WHO, IOM, and EFSA.

## Data Availability

Not applicable.

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
