# Peer review of "Thyroidal and Extrathyroidal Requirements for Iodine and Selenium: A Combined Evolutionary and (Patho)Physiological Approach"

_nutrients, 2022, doi:10.3390/nu14193886_

Round 1

Reviewer 1 Report

The Authors provide a review of thyroidal and extrathyroidal requirements for iodine and selenium, also from an evolutionary and pathophysiological approach. I really appreciated the review and I think that it could be of help and also illuminating to the scientific community. The review is extremely interesting. I have just a few observations:

-. Line 95-97: there are a series of percentages, of which the sum does not male 100%. It could be measliding, please rephrase;

- Line 548: actually, lots of reports say that, worldwide, the consumption of milk and dairy products is declining (i.e.: calcium intake trends and health consequences from childhood through adulthood, J. Am. Coll. Nutr 2003). Please take this trend into account when talking about our changing diet. Written in this way, it could be perceived that the increase in cow’s milk consumption documented in dutch women is a general trend, balancing the lower use in bakeries of iodized salt.

Author Response

Q: Line 95-97: there are a series of percentages, of which the sum does not make 100%. It could be misleading, please rephrase

'The total body iodine content is 25-50 mg, of which 50-70% is located in extra-thyroidal tissues, 60-80% is for non-hormonal use and 23% is located in the gastro-salivary pool'

Response: the reviewer is right, the percentages are mixed up and confusing. We rephrased the sentence as follows: 

'The total body iodine content is about 30–50 mg and less than 30% is present in the thyroid gland and in its hormones. Approximately 60–80% of total iodine is nonhormonal and concentrated in extrathyroidal tissues, and 23% is in the gastro-salivary pool [5-8].'

Q: Line 548: actually, lots of reports say that, worldwide, the consumption of milk and dairy products is declining (i.e.: calcium intake trends and health consequences from childhood through adulthood, J. Am. Coll. Nutr 2003). Please take this trend into account when talking about our changing diet. Written in this way, it could be perceived that the increase in cow’s milk consumption documented in Dutch women is a general trend, balancing the lower use in bakeries of iodized salt. 

Response: Yes, the effect of increased iodine levels in cow's milk is typically for the Netherlands in this case. We rephrased line 548 as follows:

'The genuine drop by this action may have been counteracted in part by higher iodide in milk, at least as observed in the Netherlands [149]. '

Reviewer 2 Report

The manuscript “Thyroidal and extra-thyroidal requirements for iodine and selenium: A combined evolutionary and (patho)physiological approach“ is interesting and well organized.

Several remarks regarding this manuscript:

1.Why did you put in brackets word (brain) in several sentences? (lines 36, 51, 330, 402). Is this tissue most related with amount and turnover of selenium and iodine in the body?

2.Numbering of sections starting with Evolutionary background (line 255) and so on must be corrected.

3.Fig. 2 is not clear enough. The time of scans must be clear and readable. Moreover, it was impossible to download and open Supplemental materials.

4.The idea of the sentence The connection between infection and the conversion of T4 to reverse T3 by DIOs 1 or 3 might aim at the feeding of the peroxidase partner system with iodide from a store. (line 378-379) is not clear and therefore must be corrected.

5.The numbers in the formula of hydrogen peroxide must be in subscript (line 796)

Author Response

Q: Why did you put in brackets word (brain) in several sentences? (lines 36, 51, 330, 402). Is this tissue most related with amount and turnover of selenium and iodine in the body?

Response: Correct, the brain tissue is of particular importance in these cases. 

Q: Numbering of sections starting with Evolutionary background (line 255) and so on must be corrected.

Response: Understood. We adjusted the numbering.

Q: Fig. 2 is not clear enough. The time of scans must be clear and readable. Moreover, it was impossible to download and open Supplemental materials.

Response: We replaced the figure with one having clear time indications. Hope this answers your question. No idea why supplemental materials could not be downloaded, but will try to solve that.

Q: The idea of the sentence The connection between infection and the conversion of T4 to reverse T3 by DIOs 1 or 3 might aim at the feeding of the peroxidase partner system with iodide from a store. (line 378-379) is not clear and therefore must be corrected.

Response: To address it more specific, the sentence has been changed into: 

'Second, the primary functions of the selenium-containing iodothyronine deiodinases (DIO1 and DIO3) might have been the release of iodide from a storage molecule, and not the synthesis and degradation of a hormone. The connection between infection and the conversion of T4 to reverse T3 by DIOs 1 or 3, resulting in the liberation of iodide from an iodide-store for its subsequent use in the anti-microbial peroxidase partner system, supports this thought.'

Q: The numbers in the formula of hydrogen peroxide must be in subscript (line 796)

Response:  We made the change and checked the document on this. 

Finally, the revised manuscript has been checked on English grammar by a native English speaking colleague